

# Rapid detection of *Enterococcus* and vancomycin resistance using recombinase polymerase amplification

Pimchanok Panpru[1], Arpasiri Srisrattakarn[1], Nuttanun Panthasri[2], Patcharaporn Tippayawat[1], Aroonwadee Chanawong[1], Ratree Tavichakorntrakool[1], Jureerut Daduang[1], Lumyai Wonglakorn[3] and Aroonlug Lulitanond[1]

[1] Centre for Research and Development of Medical Diagnostic Laboratories, Faculty of Associated Medical Sciences, Khon Kaen University, Khon Kaen, Thailand
[2] The 8th Regional Medical Science Center, Udon Thani, Thailand
[3] Clinical Microbiology Unit, Srinagarind Hospital, Khon Kaen University, Khon Kaen, Thailand

Corresponding author
Aroonlug Lulitanond,
arolul@kku.ac.th

## ABSTRACT

Vancomycin-resistant enterococci (VRE), especially *Enterococcus faecium*, have been a global concern, often causing serious healthcare-associated infections. We established a rapid approach for detecting *E. faecium* and vancomycin-resistance genes (*vanA* and *vanB*) in clinical samples using isothermal recombinase polymerase amplification (RPA) combined with a lateral-flow (LF) strip. Specific RPA primer sets and probes for *ddl* (to identify the presence of *E. faecium*) *vanA* and *vanB* genes were designed. The RPA reaction was performed under isothermal condition at 37 °C within 20 min and read using the LF strip within a further 5 min. A total of 141 positive blood-cultures and 136 stool/rectal swab samples were tested using RPA-LF method compared to the conventional PCR method. The RPA-LF method exhibited 100% sensitivity in both blood-culture (60 *E. faecium*; 35 *vanA* type and two *vanB* type) and stool/rectal-swab samples (63 *E. faecium* and 36 *vanA* type) without cross-reaction (100% specificity). The lower detection limit of the RPA-LF was approximately 10 times better than that of the conventional PCR method. The RPA-LF method is an alternative rapid method with excellent sensitivity and specificity for detecting *E. faecium*, *vanA*, and *vanB*, and it has the potential to be used as a point-of-care device for VRE therapy and prevention.

## INTRODUCTION

Enterococci are found in the gastrointestinal tract of animals and humans as well as in soil, water and food contaminated with feces. They are used as a bacterial indicator of fecal contamination in water and food. The concentration of enterococci in human stool ranged from $10^5$–$10^8$ colony forming unit (CFU)/gram (*Gelsomino et al., 2003*; *Kleessen, Bezirtzoglou & Mättö, 2000*). In addition, they are also one of the most prevalent nosocomial pathogens, causing various infections such as urinary tract infections, endocarditis and bacteremia. In the treatment of enterococcal infection, ampicillin remains the antibiotic of choice, while vancomycin is utilized in cases of ampicillin resistance

(*Kristich, Rice & Arias, 2014*). The increased usage of vancomycin has led to a rise in the number of vancomycin-resistant enterococci (VRE) infections. VRE was found to be 8.10% prevalent throughout Asia. The *Enterococcus faecium* showed higher resistance to vancomycin than *E. faecalis* (22.4% *vs.* 3.7%) (*Shrestha et al., 2021*). As shown in a survey from China, *Enterococcus* spp. was the most common pathogen in nosocomial bloodstream infections, of which 74% were *E. faecium* and 20% were *E. faecalis*. Furthermore, it was associated with high mortality rate of up to 24% (*Zhang et al., 2017*). The National Antimicrobial Resistance Surveillance Centre of Thailand revealed that VRE infection rate of *E. faecium* has climbed from 0.7% in 2012 to 6.9% in 2020, whereas VRE in *E. faecalis* has remained at 0.3% for the past ten years (*National Antimicrobial Resistance Surveillance Center Thailand, 2020*). The *vanA* and *vanB* types of VRE are the most common, especially in *E. faecium* isolates (*Hollenbeck & Rice, 2012*). The VRE is associated with high-mortality enterococcal bloodstream infections (*Chiang et al., 2017*). Most hospitals screen for VRE carriers from rectal swab samples, in order to avoid enterococcal bloodstream infections, prevent the spread of nosocomial infection and establish a successful treatment strategies.

The conventional identification of VRE is generally based on culture, biochemical tests, disk diffusion and determination of minimal inhibitory concentration, as well as genotypic methods (*Jenkins & Schuetz, 2012*). However, all these methods are time-consuming. Molecular techniques were developed to identify the species of *Enterococcus* several decades ago (*Dutka-Malen, Evers & Courvalin, 1995*). Although the polymerase chain reaction (PCR) method is highly sensitive and specific, it has been limited to well-equipped facilities due to the thermocycling requirements. The isothermal amplification techniques have been developed to overcome this constraint, such as recombinase polymerase amplification (RPA), which was initially reported in 2006 (*Piepenburg et al., 2006*). RPA technique is based on the functions of three proteins comprising recombinase, recombinase loading factor and single-stranded binding (SSB) protein. The nucleoprotein complex is formed by the interaction of an oligonucleotide primer and a recombinase. The recombinase loading factor assists in the combination. The complex searches for the homologous sequences on DNA template and replaces them in the duplex DNA. The formation of primer-DNA complex is stabilized by SSB protein (*Yonesaki & Minagawa, 1985*). DNA polymerase initiates the amplification at the end of the primer and finally results in the amount of amplified DNA (*Piepenburg et al., 2006*). The RPA reaction is enzymatically processed and does not require thermal cycling. The RPA technique can detect very small amount of target DNA molecules. It only requires a low constant temperature (25–42 °C) and a 20 min turnaround time. Moreover, the RPA product can be detected by using a variety of methods, such as agarose gel electrophoresis (AGE), real-time quantitative fluorescence or electrochemical detection, etc. Alternatively, the lateral-flow strips (LF) can be used to visualize the amplicons in 5 min, which is highly practical. The LF strip is a simple tool for qualitative testing, based on sandwiches assay by adding a specific probe into RPA reaction solution. The RPA products would be detected by using specific antibody to the pathogen, tagged with gold nanoparticles which is applied on a paper strip. The results of LF are visible and can be read in 5–10 min (*Jauset-Rubio et al., 2016*).

Therefore, we developed a rapid and sensitive RPA-LF method for detecting *E. faecium* and its important vancomycin-resistance genes (*vanA* and *vanB*) in positive blood-culture, stool and rectal-swab samples.

## MATERIALS & METHODS

### Bacterial isolates and clinical samples

Ethical approval for this study was obtained from Human Ethics Committee of Khon Kaen University (HE611605). A total of 124 bacterial isolates and 25 clinical samples were collected from patients of a university hospital between January 2018 and December 2020. The 124 bacterial isolates consisted of 20 *E. faecium* (not VRE), 35 *vanA*-carrying *E. faecium,* 48 *E. faecalis*, five *Enterococcus* spp. and 16 non-*Enterococcus* isolates. The 25 clinical samples were 15 positive blood-culture samples (four *E. faecium*, five *E. faecalis*, one *E. casseliflavus,* and five non-*Enterococcus*) and 10 rectal swab samples for VRE screening (six *E. faecium* (not VRE), one *vanA*-carrying *E. faecium,* one *E. faecalis*, and two group D streptococci). Additionally, two reference strains, *vanB*-carrying *E. faecium* and *vanB*-carrying *E. faecalis*, were purchased from the Department of Medical Sciences, Ministry of Public Health (Nonthaburi Province, Thailand). All isolates in the Table 1 were identified by using biochemical tests such as bile esculin, 6.5% sodium chloride (NaCl), arabinose, sorbital and motility tests (*Teixeira et al., 2011*), then *E. faecium* was confirmed by conventional PCR amplification of the *ddl* gene. In addition, *vanA* and *vanB* genes were detected by conventional PCR methods. The isolates were stored at −20 °C in skimmed milk plus 20% glycerol. Before testing, all isolates were sub-cultured on blood agar and incubated overnight at 37 °C. Fresh colonies were suspended in 50 µL of sterile distilled water then put in boiling water for 10 min. After centrifugation at 13,500 rpm for 1 min, the supernatant was used as the DNA template for either conventional PCR or RPA reaction. All isolates were used to spike in blood and stool samples as well.

### Identification of *E. faecium* (*ddl* gene) and VRE (*vanA* and *vanB* genes) by conventional PCR

To identify *E. faecium* and *vanA* or *vanB* genotypes of VRE, a region of the *ddl*, and of the *vanA* and *vanB* genes were amplified by PCR reaction using our designed RPA primer set for *ddl* [ddl-F(RPA) and ddl-R(RPA)], while primers used for *vanA* and *vanB* genes were according to a previous report (Table 2) (*Dutka-Malen, Evers & Courvalin, 1995*). The 25 µL PCR reaction contained 0.2 µM of each primer, 1X PCR buffer, 0.2 mM dNTP, 2 mM MgCl$_2$, 1 U *Taq* polymerase and 3 µL of DNA template. The PCR cycling conditions were 94 °C for 2 min, followed by 30 cycles of 94 °C for 1 min, 54 °C for 1 min, and 72 °C for 1 min, with a final step at 72 °C for 10 min. The PCR products were analyzed using 2% agarose gel electrophoresis (AGE), stained with ethidium bromide solution and visualized under UV transilluminator.

### Design of primers and probes for screening

The single-stranded RPA primers and probes specific for the *ddl* gene of *E. faecium*, *vanA* and *vanB* genes were designed by using the ClustalW program to align consensus sequences

Panpru et al. (2021), PeerJ, DOI 10.7717/peerj.12561

Peer J

**Table 1** Results of PCR-AGE and RPA-LF methods for detection of *E. faecium*, *vanA* and *vanB* genes.

| Organisms (n) | Number of positive result | | | | | | | |
|---|---|---|---|---|---|---|---|---|
| | *E. faecium* | | | *vanA* | | | *vanB* | |
| | PCR-AGE[a] | RPA-LF[b] | RPA-LF[c] | PCR-AGE[a] | RPA-LF[b] | RPA-LF[c] | PCR-AGE[a] | RPA-LF[d] |
| **Bacterial isolates (126)** | | | | | | | | |
| *E. faecium* (20) | 20 | 20 | 20 | 0 | 0 | 0 | 0 | 0/6 |
| *vanA*-carrying *E. faecium* (35) | 35 | 35 | 35 | 35 | 35 | 35 | 0 | 0/35 |
| *vanB*-carrying *E. faecium* (1) | 1 | 1 | 1 | 0 | 0 | 0 | 1 | 1/1 |
| *E. faecalis* (48) | 0 | 0 | 0 | 0 | 0 | 0 | 0 | 0/5 |
| *vanB*-carrying *E. faecalis* (1) | 0 | 0 | 0 | 0 | 0 | 0 | 1 | 1/1 |
| *E. gallinarum* (2) | 0 | 0 | 0 | 0 | 0 | 0 | 0 | 0/2 |
| *E. casseliflavus* (1) | 0 | 0 | 0 | 0 | 0 | 0 | 0 | 0/1 |
| *E. raffinosus* (1) | 0 | 0 | 0 | 0 | 0 | 0 | 0 | 0/1 |
| *E. avium* (1) | 0 | 0 | 0 | 0 | 0 | 0 | 0 | 0/1 |
| *S. agalactiae* (5) | 0 | 0 | 0 | 0 | 0 | 0 | 0 | ND |
| *S. pyogenes* (3) | 0 | 0 | 0 | 0 | 0 | 0 | 0 | 0/1 |
| *S. dysgalactiae* (1) | 0 | 0 | 0 | 0 | 0 | 0 | 0 | ND |
| *S. constellatus* (1) | 0 | 0 | 0 | 0 | 0 | 0 | 0 | ND |
| *S. aureus* (1) | 0 | 0 | 0 | 0 | 0 | 0 | 0 | 0/1 |
| *S. haemolyticus* (1) | 0 | 0 | 0 | 0 | 0 | 0 | 0 | ND |
| *E. coli* (1) | 0 | 0 | 0 | 0 | 0 | 0 | 0 | 0/1 |
| *K. pneumoniae* (1) | 0 | 0 | 0 | 0 | 0 | 0 | 0 | 0/1 |
| *A. baumannii* (1) | 0 | 0 | 0 | 0 | 0 | 0 | 0 | 0/1 |
| *P. aeruginosa* (1) | 0 | 0 | 0 | 0 | 0 | 0 | 0 | 0/1 |
| **Positive blood culture (15)** | | | | | | | | |
| *E. faecium* (4) | 4 | 4 | ND | 0 | 0 | ND | 0 | 0/4 |
| *E. faecalis* (5) | 0 | 0 | ND | 0 | 0 | ND | 0 | 0/5 |
| *E. casseliflavus* (1) | 0 | 0 | ND | 0 | 0 | ND | 0 | 0/1 |
| *S. aureus* (1) | 0 | 0 | ND | 0 | 0 | ND | 0 | 0/1 |
| *S. epidermidis* (1) | 0 | 0 | ND | 0 | 0 | ND | 0 | 0/1 |
| Coagulase negative staphylococci (1) | 0 | 0 | ND | 0 | 0 | ND | 0 | 0/1 |
| *S. viridans* (1) | 0 | 0 | ND | 0 | 0 | ND | 0 | 0/1 |
| *S. gallolyticus* (1) | 0 | 0 | ND | 0 | 0 | ND | 0 | 0/1 |
| **Rectal swab (10)** | | | | | | | | |
| *E. faecium* (6) | 6 | ND | 6 | 0 | ND | 0 | ND | ND |
| *vanA*-carrying *E. faecium* (1) | 1 | ND | 1 | 1 | ND | 1 | ND | ND |
| *E. faecalis* (1) | 0 | ND | 0 | 0 | ND | 0 | ND | ND |
| group D streptococci non-enterococcus (2) | 0 | ND | 0 | 0 | ND | 0 | ND | ND |

Panpru et al. (2021), *PeerJ*, DOI 10.7717/peerj.12561

**Table 1** (*continued*)

| Organisms (n) | Number of positive result | | | | | | | |
|---|---|---|---|---|---|---|---|---|
| | *E. faecium* | | | *vanA* | | | *vanB* | |
| | **PCR-AGE[a]** | **RPA-LF[b]** | **RPA-LF[c]** | **PCR-AGE[a]** | **RPA-LF[b]** | **RPA-LF[c]** | **PCR-AGE[a]** | **RPA-LF[d]** |
| Total (151) | 67 | 60 | 63 | 36 | 35 | 36 | 2 | 2/74 |

**Notes.**

ND, not done.

[a]The conventional PCR method was tested using DNA from bacterial colonies.

[b]RPA-LF method for *E. faecium* and *vanA* was tested in positive blood culture samples.

[c]RPA-LF method for *E. faecium* and *vanA* was tested in stool/rectal swab samples.

[d]RPA-LF method for *vanB* was tested in 74 positive blood culture samples.

Panpru et al. (2021), *PeerJ*, DOI 10.7717/peerj.12561

**Table 2  Single strand primers and probes used for PCR and RPA reactions.**

| Target genes | Primer name | Sequence (5′–3′) and modification | Product length | Primer location | Genbank accession no. | Reference |
|---|---|---|---|---|---|---|
| *ddl* | *ddl*-F (RPA)[a] <br> *ddl*-R (RPA)[a] | ACCCAAGTGGACAGACAGAGGAAGGCTTTA <br> TTCCATCTTCCCCGTTTGGCCCATGTAAAACT | 156 bp | 242–271 <br> 368–397 | AY489046.1 | This study |
| | *ddl*-R (RPA-LF) <br> *ddl*-P (RPA-LF) | Biotin-CATAAGGCATATTCAATGTCTCTAAGAAGC <br> FAM-CGGGAGAAATCAAAGAAGAAGGAGCCATCG-[THF]-TTTTCCAGTTTTACA-C3-Spacer | | | | This study |
| *vanA* | *vanA*-F (PCR) <br> *vanA*-R (PCR) | GGGAAAACGACAATTGC <br> GTACAATGCGGCCGTTA | 941 bp | 176–192 <br> 891–907 | NC_014475.1 | *Dutka-Malen, Evers & Courvalin (1995)* |
| | *vanA*-F (RPA) <br> *vanA*-R (RPA) | TTGCGCGGAATGGGAAAACGACAATTGCTATT <br> CAAAAGGGATACCGGACAATTCAAACAGACC | 194 bp | 165–196 <br> 328–358 | NC_014475.1 | This study |
| | *vanA*-R (RPA-LF) <br> *vanA*-P (RPA-LF) | Biotin-CAAAAGGGATACCGGACAATTCAAACAGACC <br> FAM-GATGTAGCATTTTCAGCTTTGCATGGCAAG-[THF]-CAGGTGAAGATGGAT-C3-Spacer | | | | This study |
| *vanB* | *vanB*-F (PCR) <br> *vanB*-R (PCR) | ATGGGAAGCCGATAGTC <br> GATTTCGTTCCTCGACC | 635 bp | 174–190 <br> 792–808 | NC_004668.1 | *Dutka-Malen, Evers & Courvalin (1995)* |
| | *vanB*-F (RPA) <br> *vanB*-R (RPA) | GAGGATGATTTGATTGTCGGCGAAGTGGAT <br> TTTGCCGTTTCTTGCACCCGATTTCGTTCCTC | 165 bp | 664–693 <br> 765–795 | NC_004668.1 | This study |
| | *vanB*-R (RPA-LF) <br> *vanB*-P (RPA-LF) | Biotin- TTTGCCGTTTCTTGCACCCGATTTCGTTCCTC <br> FAM-CAAATCCGGCTGAGCCACGGTATCTTCCGC-[THF]-TCCATCAGGAAAACG-C3-Spacer | | | | This study |

**Notes.**

[a]The *ddl* RPA primer pair was used in both PCR and RPA reaction.

F, forward primer; R, reverse primer; FAM, 6-Carboxyfluorescein; THF, tetrahydrofuran residue, an internal abasic nucleotide; C3-Spacers, a polymerase extension blocking group.

into multiple-sequence alignment. Primers and probes were manually designed according to the guidelines of the manufacturer (TwistDX, Cambridge, UK) and the %GC content, secondary structure, potential for primer-dimer and hairpin formation were checked using OligoEvaluator™ and OligoAnalyzer™. The possibility of false priming with other related genes was examined by using the Primer-BLAST program of the National Center for Biotechnology Information. PCR-AGE and RPA-AGE techniques were used to identify the most appropriate primers for the target amplification.

The final sequences of RPA primers and probes are listed in Table 2. To enable the lateral flow detection, the reverse primer was conjugated with biotin at the 5′ end and the probe (46–52 bp) was designed with a 5′ antigenic label (FAM), an internal abasic nucleotide (tetrahydrofuran residue or THF) and a polymerase extension blocking group at the 3′ end (*TwistDx™ Limited, 2018*).

## RPA reaction components

The RPA-AGE reaction was performed using TwistAmp Basic kit (TwistDX, Cambridge, UK). RPA components were mixed according to a previous study (*Srisrattakarn et al., 2020*). The RPA reaction was incubated at 37 °C for 20 min and the reaction was stopped by heating at 65 °C for 10 min to eliminate the excess protein (*Kapoor et al., 2017*). The RPA products were then diluted (1:1) in deionized water before AGE.

The RPA-LF reaction was performed using TwistAmp nfo kits (TwistDx, Cambridge, UK). The RPA master mix contained 2.1 μL of each primer (10 μM), 29.5 μL of 1× rehydration buffer, 0.6 μL of probe (10 μM), and 11.2 μL of deionized water. A total of 45.5 μL of master mix were transferred to a dried enzyme pellet tube and mixed well by pipetting. The mixture was divided into four aliquots (11.37 μL each) in 0.2 mL tubes. DNA template or sample (0.5 μL) was added to each tube. The reaction was initiated by the addition of 0.63 μL of 280 mM magnesium acetate. The lateral-flow strip (Milenia Genline HybriDetect-1; TwistDx, Cambridge, UK) was used to detect the RPA product by dipping the strip into a mixture of 0.5 μL of RPA product and 50 μL of phosphate-buffered saline with Tween at room temperature. The result was read within 5 min. The presence of a control band helps to validate successful test: a clearly visible band at both the control and test lines indicate a positive result and the appearance of only a control band indicates a negative result.

## Optimization for RPA reaction

The optimum temperature and incubation time for the RPA reaction was tested by RPA-AGE and RPA-LF methods. DNA extracted from $10^8$ CFU/mL of bacterial suspension was used as template for the RPA reactions. Blank control tube was tested together with each experiment. The main parameters of RPA are amplification temperature and reaction time. The optimal conditions of RPA (for all three genes) were found by trying various temperatures (25, 28, 31, 34, 37 and 42 °C) and various reaction times (5, 10, 15, 20, 25 and 30 min). The final step of all RPA reactions was 65 °C for 10 min. Before incubation at various temperatures, each reaction tube was transported in a PCR-cooler tube rack. The optimum reaction was determined by the most obvious color on the detection line by the naked eye or the specific band in agarose gel.

## The detection limit of PCR-AGE, RPA-AGE and RPA-LF methods

The detection limits of the three methods were determined by using a 10-fold serial dilution ($10^{-1}$–$10^{-7}$) of DNA extracted from a bacterial suspension (turbidity to 3 McFarland standard) as templates for PCR and RPA reactions. The amplicons were analyzed using AGE and LF methods. Meanwhile, the number of bacteria in each suspension was determined in duplicate by using a spread plate technique. The bacterial colonies were counted and CFU/mL calculated. The lowest quantity of DNA template that yielded a positive result was considered to be the detection limit. Each dilution was tested in duplicate.

## Performance of RPA-LF for testing clinical samples
### Positive blood-culture samples

Each of the 126 bacterial isolates was prepared to a suspension of $10^5$ CFU/mL in normal saline solution. Spiked blood-culture samples were prepared by mixing 500 μL of the bacterial suspension with 10 mL of healthy human blood and transferred to pre-incubated aerobic culture bottles (RENDER; Zhuhai Meihua Medical Technology Limited, Zhuhai, China) to make a final inoculum of approximately 500 CFU/mL (*Srisrattakarn et al., 2020*) and incubated overnight at 37 °C. The number of bacteria after positive signal were approximately $10^8$–$10^9$ CFU/mL (*Smith, 2018*). Additionally, 15 positive blood-culture bottles, identified as positive by the BacT/Alert® Virtuo Microbial Detection System (bioMérieux, Marcy l'Etoile, France), from Srinagarind Hospital (a university hospital, Northeast of Thailand) were also tested. All the 141 samples were directly tested using the RPA-LF method for presence of *E. faecium* and of the *vanA* gene, while 74 samples were tested for the *vanB* gene.

### Stool samples

Spiked stool samples were prepared by mixing 10 μL of each of 126 bacterial isolates with approximately 0.1 g of stool. The samples were then suspended in 990 μL of bile esculin broth (BE broth) for enrichment of *Enterococcus*. After incubated overnight at 37 °C, the colony count was performed from two representative samples which found approximately $10^8$ CFU/mL. The samples were centrifuged at 3,000 rpm for 5 min to remove stool debris. The supernatant was further centrifuged at 13,000 rpm for 10 min and all liquid was removed. The microbial sediment was resuspended in distilled water and directly used in the RPA-LF reaction for detection of *E. faecium* and the *vanA* gene. Ten additional rectal-swab samples from VRE-screened patients attending Srinagarind Hospital were also tested in the same manner.

The performance of the RPA-LF method was evaluated in clinical samples compared with the PCR-AGE method. The results of both methods were blindly read and recorded by three independent research technicians. The final result was reported based on agreement by at least two of the three readers. The sensitivity, specificity, and 95% confidence interval (CI) of the RPA-LF method were calculated by using a free software VassarStats (http://vassarstats.net/) (*Dortet et al., 2018*).

## RESULTS

### Optimal temperature and time for RPA reaction

The RPA-AGE method operated well at a wide range of temperatures from 25–42 °C for the detection of *ddl*, *vanA* and *vanB* genes (Fig. S1), whereas the RPA-LF method for the *ddl* gene provided a strong test-line band at temperatures ranging from 28–42 °C (Fig. S2). The optimum temperature for the further test was 37 °C. Both RPA-AGE and RPA-LF methods performed at 37 °C yielded an obvious band at the period ranging from 10-30 min for detection of the three genes (Figs. S1 and S2, respectively). All blank control tubes in each condition provided negative results. Finally, the optimal condition of RPA-LF reaction at 37 °C for 20 min was selected for the subsequent testing.

### Determination of the detection limit of PCR-AGE, RPA-AGE and RPA-LF method

The lower limit of the PCR-AGE method for detection of *E. faecium*, and of *vanA*- and *vanB*-carrying VRE were $10^8$, $10^6$ and $10^7$ CFU/mL, respectively, while the detection limit of the RPA-LF and RPA-AGE methods were $10^7$, $10^5$ and $10^6$ CFU/mL, respectively (Fig. 1). The detection limit of RPA method was 10 times lower than that of the PCR-AGE method.

### Detection of bacterial isolates in clinical samples by RPA-LF

Using the PCR-AGE as the reference method, the detection of *E. faecium*, and of *vanA* and *vanB* genes in positive blood-culture samples by the RPA-LF method gave sensitivity and specificity of 100% (*E. faecium,* 60/60, 81/81; *vanA*, 35/35, 106/106; *vanB*, 2/2, 72/72 for positive and negative samples, respectively). Similarly, the detection of these targets in stool/rectal swab samples showed congruent results with the PCR-AGE method. The sensitivity and specificity of the RPA-LF method were 100% (*E. faecium,* 63/63, 73/73; *vanA,* 36/36, 100/100 for positive and negative samples, respectively) (Table 1 and Table S1).

## DISCUSSION

*E. faecium* is microorganism present in the intestines of humans and animals that can cause various infections including bacteremia, which can lead to considerable morbidity and mortality. In the United States, around 83 percent of *E. faecium* were vancomycin resistant in bloodstream infections patients (*Weiner et al., 2016*). VRE infections in the bloodstream had a greater mortality rate than those of vancomycin-susceptible *Enterococcus* (*DiazGranados et al., 2005*). VRE can survive in various adverse environments such as a wide range of temperature (10–45 °C) and pH (pH 4.6–9.9), 40% bile salts and 6.5% NaCl (*Vanden Berghe, De Winter & De Vuyst, 2006*; *Foulquie Moreno et al., 2006*), particularly in healthcare settings. VRE-carrier screening using stool or rectal swab sample is recommended by Hospital Infection Society and Infection Control Nurses Association to reduce the spread of VRE and related infections in hospitals (*Cookson et al., 2006*). Development of reliable and sensitive methods for VRE detection is therefore of paramount importance. The RPA-LF method has been adapted to detect many pathogens which requires minimal equipment and a quick turnaround time of 30 min. It has been applied for detection of pathogens in various samples, such as blood

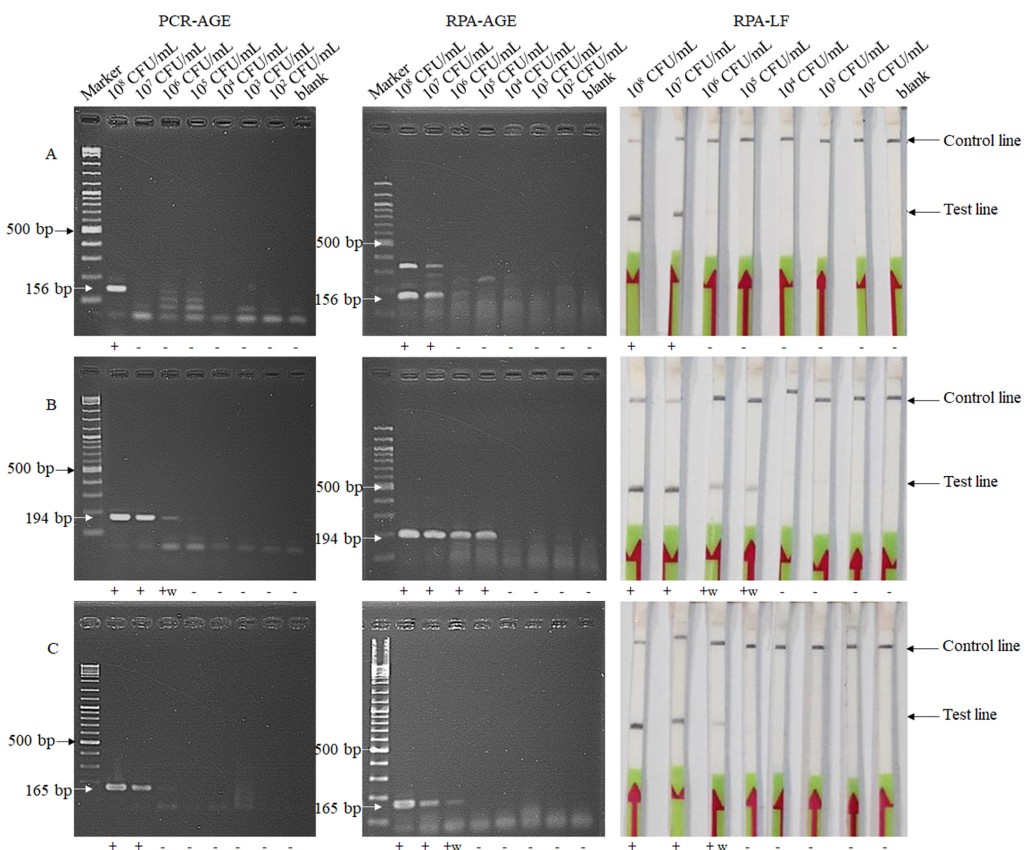

**Figure 1  Detection limits of the RPA-AGE and RPA-LF methods compared with those of the PCR method for detection of *E. faecium* (A), *vanA* (B) and *vanB* (C) genes.** Each sample gave the same result after being tested twice. Marker, GeneRuler DNA Ladder Mix (Thermo Scientific, Waltham, MA, USA); +, positive result; +w, weakly positive result; -, negative result.

(*Srisrattakarn et al., 2020*), plasma (*Qi et al., 2018*), foodstuffs (*Du et al., 2018a*; *Du et al., 2018b*), stool (*Crannell et al., 2014a*) and environmental samples such as plant (*Londono, Harmon & Polston, 2016*), soil and water (*Saxena et al., 2019*). To the best of our knowledge, this is the first report of the application of the RPA-LF method for direct detection of *E . faecium* and the *vanA* and *vanB* genes in positive blood-culture bottles and stool/rectal-swab samples, facilitating immediate medical intervention.

The optimal conditions of the RPA reaction for detection of the three targets (*E . faecium* and *vanA* and *vanB* genes) were 37 °C for 20 min. Several investigations have shown that the RPA-LF can function well at temperatures ranging from 25–45 °C (*Lu et al., 2021*; *Peng et al., 2019*) and the faint band was visible in 5 min (*Du et al., 2018a*; *Peng et al., 2019*; *Srisrattakarn et al., 2020*; *Wu et al., 2020*). Similarly, our study found that the RPA worked effectively at a wide temperature range (25–42 °C) and incubation times of 5-30 min could generate a visual signal on the test lines. We chose 37 °C because all reactions yielded strong bands at the test lines and because low-resource settings usually have equipment allowing incubation at this temperature. The RPA reaction was incubated for 20 min to prevent false

positive results due to unexpected amplification in long time of reaction (*Piepenburg et al., 2006*). Note that the RPA product was diluted 1:100 with buffer to reduce nonspecific binding of crowding agent and proteins in RPA reactions with the antibodies on a lateral flow strip, and the LF test must be read within 5 min to prevent a false positive result (*Rosser et al., 2015*; *Saldarriaga et al., 2016*; *Wu et al., 2016*).

The RPA-LF method for detection of the three genes was 10 times more sensitive (lower detection limit) than the conventional PCR method, whereas the detection limits of the RPA-LF and RPA-AGE methods were equal. The RPA reaction has higher sensitivity than the PCR method because the RPA primer is designed according to the TwistAmp® kit formulations which indicated the support to the RPA reaction. The RPA reactions contain a high level of ATP-burning recombinase that the fuel is consumed typically within around 25-30 min. RPA primers (30–35 bp) are longer than the typical conventional PCR primers for the amplification to give a DNA product of 100–500 bp (*TwistDx Limited, 2018*). The primers length of greater than 28 bp bind strongly to the recombinase protein. These complex searches for the homologous sequences in the double-stranded DNA and invading the DNA template. The recombinase-primer complex stimulates the ATP hydrolysis in RPA reaction. The complex will split when the ATP is hydrolyzed, and the recombinase will bind to the primer to begin the next reaction immediately (*Piepenburg et al., 2006*). Therefore, the RPA has higher sensitivity and gives a faster result than the conventional PCR reaction. The detection limit of RPA in our study was lower than the conventional PCR, which was consistent with the study of *Wang et al. (2016)*.

The performance of the RPA-LF for detection of pathogens in both blood culture and rectal swab/stool samples showed 100% sensitivity and 100% specificity with no cross reactions. The method has been successful for direct detection of *E . faecium* and *vanA* or *vanB* genes in positive blood-culture samples. Several studies showed that some components of blood, mainly heme (*Akane et al., 1994*), leukocyte DNA (*Morata, Queipo-Ortuno & Colmenero, 1998*), immunoglobulin G in plasma (*Al-Soud, Jonsson & Radstrom, 2000*), and various inhibitors in the environment (*Jiang et al., 2005*) are major inhibitors of the PCR reaction. The RPA-LF technique was shown to be successful even in serum, heparin and hemoglobin (*Kersting et al., 2014*). However, some technical problems of the RPA-LF reaction were encountered during the study, when tested with fecal samples directly. It is known that fecal samples contain various substances that influence the reactions (*Schrader et al., 2012*). Additionally, the culture medium for bacterial enrichment or the solution for sample extraction may contain inhibitors. In this study, the bile esculin inhibited the RPA-LF reaction resulting in negative results (Fig. S3). The cetyltrimethyl ammonium bromide in DNA extraction buffer was reported to have a strong inhibitory effect on RPA reactions (*Valasevich & Schneider, 2017*). Therefore, sample preparation steps are crucial to reduce those substances. In our study, the faecal debris and medium broth were eliminated as feasible after the incubation period. Then the suspected microbial suspension was used as sample for the RPA-LF reaction, which yielded satisfactory results despite the lack of DNA extraction. The RPA-LF reaction can tolerate crude samples with minimal sample preparation steps (*Silva et al., 2018*) and in the presence of many inhibitors (*Kersting et al., 2014*). In comparison to the results obtained from conventional PCR method and the

developed RPA method of *Yin et al. (2017)*, the RPA method has a high sensitivity and specificity. However, development of RPA for detection of *Burkholderia pseudomallei* in blood samples (*Peng et al., 2019*) and detection of group B streptococci in vaginal/anal samples (*Daher et al., 2014*) were compared to real-time PCR method, the latter report showed that RPA assay had 96 and 100% of sensitivity and specificity, respectively.

In our detection limit study, the result of conventional PCR reaction was negative, while that of the RPA reaction was positive. This occurred when there was low number of bacteria in the sample which affected the capability of the detection method. The number of bacteria in each sample in this study was at least $10^8$ CFU/ mL, which could be detected from both RPA and conventional PCR. As a result, the calculated values of sensitivity and specificity were 100%. However, the real-time PCR method should be used as a reference method to support in the further research, particularly for the detection of samples containing a low number of bacteria.

The RPA method uses simple incubation conditions and requires common laboratory equipment such as a heat block and water bath. Moreover, the RPA processes for DNA amplification can be incubated at the body temperature (*Crannell, Rohrman & Richards-Kortum, 2014b*). While the conventional or real-time PCR methods and GeneXpert need special equipment such as thermocycle machine or GeneXpert instrument (Cepheid, Sunnyvale, CA, USA). RPA method displays a faster time-to-result (<20 min) than the PCR method (>1.5 hr) and the GeneXpert (~45 min) (*Moore & Jaykus, 2017*; Cepheid, Sunnyvale, CA, USA). These are the main benefits of the RPA for VRE detection at the point of care setting. Furthermore, one of the most important factors to consider is the cost. The RPA platform is more cost-effective (~$3.70) compared with the costs of conventional PCR (~$2.60) and real-time PCR (~$25) (*Al-Siyabi et al., 2013*; *Richards et al., 2019*; *Zhang et al., 2014*). However, the volume of RPA reaction can be reduced to 12.5 µL leading to decrease in the cost of RPA-LF method to ~$2.28 per test (~$0.83 for RPA reaction and ~$1.45 for the haft of commercial LF strip), which was adopted in this study.

One limitation of this study is the small number of clinical samples that we used. VRE bloodstream infections are rare in the hospital, limiting the availability of such samples. Therefore, additional tests were carried out using spiked samples. However, additional clinical samples for testing are still needed, especially those of the *vanB*-type VRE, which are particularly uncommon in our area. In addition, *E. faecalis* can cause a variety of nosocomial infections such as urinary tract infections but the prevalence of vancomycin-resistant *E. faecalis* (0.3%) was lower than *E. faecium* (6.9%) in our area (*National Antimicrobial Resistance Surveillance Center Thailand, 2020*). Therefore, the method for detection of *E. faecalis* should be further studied. Another limitation in this study was a shortage supply of TwistAmp nfo kit from the manufacturer, forcing us to test a limited number of samples, especially for the *vanB* gene and lacking of the RPA-LF test for *E. faecalis*.

## CONCLUSIONS

The RPA-LF method is rapid (within 30 min), user-friendly test for detecting *E. faecium* and *vanA* or *vanB* genes. It exhibits high sensitivity and specificity. This is a suitable candidate

for use in low-resource laboratories and would also be useful for infection-control purposes to prevent the spread of VRE in hospitals.

## ACKNOWLEDGEMENTS

We are grateful to Assoc. Prof. Chotechana Wilailuckana, Centre for Research and Development of Medical Diagnostic Laboratories, Faculty of Associated Medical Sciences, Khon Kaen University, for kindly providing the 20 *vanA*-carrying *E. faecium* isolates; and staff of Clinical Microbiology Laboratory, Srinagarind Hospital, Khon Kaen, Thailand, for collecting the clinical isolates. We would like to acknowledge Prof. David Blair, for editing the MS *via* Publication Clinic KKU, Thailand.

### Funding

This study was supported by the KKU Research Fund (Project no. I62-00-19-03). Pimchanok Panpru was awarded by Graduate School, Khon Kaen University Research Grant (Grant no. 611JH107). The funders had no role in study design, data collection and analysis, decision to publish, or preparation of the manuscript.

### Grant Disclosures

The following grant information was disclosed by the authors:
KKU Research Fund: Project no. I62-00-19-03.
Graduate School, Khon Kaen University Research Grant: 611JH107.

### Competing Interests

The authors declare there are no competing interests.

### Author Contributions

- Pimchanok Panpru conceived and designed the experiments, performed the experiments, analyzed the data, prepared figures and/or tables, authored or reviewed drafts of the paper, and approved the final draft.
- Arpasiri Srisrattakarn conceived and designed the experiments, performed the experiments, analyzed the data, authored or reviewed drafts of the paper, and approved the final draft.
- Nuttanun Panthasri analyzed the data, authored or reviewed drafts of the paper, sample collecting, and approved the final draft.
- Patcharaporn Tippayawat, Aroonwadee Chanawong, Ratree Tavichakorntrakool and Jureerut Daduang analyzed the data, authored or reviewed drafts of the paper, and approved the final draft.
- Lumyai Wonglakorn analyzed the data, authored or reviewed drafts of the paper, sample collecting, and approved the final draft.
- Aroonlug Lulitanond conceived and designed the experiments, analyzed the data, prepared figures and/or tables, authored or reviewed drafts of the paper, and approved the final draft.

## Ethics

The following information was supplied relating to ethical approvals (i.e., approving body and any reference numbers):

Ethical approval for this study was obtained from Human Ethics Committee of Khon Kaen University (HE611605).

## Data Availability

The raw data are available in the Supplementary Files.

## Supplemental Information

Supplemental information for this article can be found online at http://dx.doi.org/10.7717/peerj.12561#supplemental-information.

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
