# Peer review of "Rapid detection of Enterococcus and vancomycin resistance using recombinase polymerase amplification"

_PeerJ, doi:10.7717/peerj.12561_

## Round 0.1 · original submission · Major Revisions

Daer Dr. Panpru,

Both expert reviewers liked the main concept of your work but suggested a series of improvements which have to be made prior to resubmission of the manuscript. I am looking forward to receiving your carefully revised work.

Best regards,
Elisabeth Grohmann

·

Basic reporting

Please see the attached PDF file.

Experimental design

Please see the attached PDF file.

Validity of the findings

Please see the attached PDF file.

Reviewer 2 ·

Basic reporting

- The manuscript contains many linguistics errors and should be evaluated by a native English speaker

Experimental design

- The validation of the newly developed test is mostly performed on spiked samples. The authors should prove that the concentrations that were used to mimic real samples are representative. Is this based on literature data?
- The authors compared the performance characteristics of the newly developed test to that of a conventional PCR and not a real-time PCR. This is a major limitation with regard to sensitivity and should be elaborated on in the discussion. Furthermore, the statements throughout the manuscript about high sensitivity and specificity should be adapted accordingly.
- The specificity of the newly developed test is 100% compared to a PCR that is less sensitive. The difference in sensitivity implies that certain samples are positive with the new test and negative with the PCR which would result in false positive results and thus a lower specificity? The authors should clarify how the specify was calculated because it cannot be the PCR that was used as reference.
- Why was analyses of external quality control samples for detection of vanA and vanB genes not part of this study, they are available and would make it possible to compare the performance of this newly developed assay

Validity of the findings

- The authors should elaborate in the discussion on an explanation for the higher sensitivity of the isothermal reaction compared to the PCR
- The authors should make a statement about the use of this test compared to commercial assays, especially the vanA/B GeneXpert which does not require specialized equipment.

Additional comments

- A ddl gene is also present in E. faecalis strains. The authors should clarify why targeting this gene in their assay only detect E. faecium strains.
- Line 97: group D streptococci is a confusing term as both E. faecium and E. faecalis belong to this group
- Line 48: the statement on increased ‘serious problems’ with VRE should be clarified or assigned to specific regions, this is not a worldwide phenomenon. Furthermore ‘serious problems’ is rather vague.
- Line 100: which conventional methods are referred to?

---

## Round 0.2 · Minor Revisions

Dear Dr. Panpru,

Your manuscript has considerably improved by the in-depth revision you performed. However, there are still several issues which have to be considered:

- Throughout the manuscript: All bacterial species have to be italicised, all gene names as well.

- Did you use double-stranded primers in your work? You give the primer lengths in bp. Please clarify.

- The English grammar is still not correct in several sections of the manuscript: Please double-check the correct use of singular and plural forms of nouns with the correct corresponding verb.

- In addition, e.g. throughout the manuscript you write PCR-base methods. I assume that you refer to PCR-based methods.

- Several non-scientific terms are used, such as "hard environment". Please be precise and specify.

Kind regards,
Elisabeth Grohmann

·

Basic reporting

no comment

Experimental design

no comment

Validity of the findings

The revised manuscript is much improved.

Additional comments

The supplemental materials, Figure S2 and Figure S3 and the quote in parentheses at line 209 in text, could be removed, because those are negative data and not needed.
Those are just necessary for this peer review only.

---

## Round 0.3 · accepted · Accept

Now all my issues have been fixed.